# Push and Pull Factors: Contextualising Biological Maturation and Relative Age in Talent Development Systems

**DOI:** 10.3390/children10010130

**Published:** 2023-01-09

**Authors:** Liam Sweeney, Jamie Taylor, Áine MacNamara

**Affiliations:** 1School of Health and Human Performance, Faculty of Science and Health, Dublin City University, Glasnevin, D09 W6Y4 Dublin, Ireland; 2Grey Matters Performance Ltd., Stratford Upon Avon CV37 9TQ, UK; 3Insight SFI Centre for Data Analytics, Dublin City University, Glasnevin, D09 W6Y4 Dublin, Ireland

**Keywords:** talent development, talent identification, bio-banding, relative age effect, talent development systems, sport performance, challenge

## Abstract

In this conceptual paper, we contextualise ongoing attempts to manage challenge dynamics in talent systems in sport. Firstly, we review the broad literature base related to biological maturation, relative age, and the proposed interventions to mitigate effects. We suggest that the relative age effect may be a population level effect, indicative of deeper phenomena, rather than having a direct effect on challenge levels. In contrast, we suggest that biological maturation has a direct effect on challenge at the individual level. Therefore, our main critique of many existing approaches to the management of challenge is a lack of individual nuance and flexibility. We suggest the necessity for talent systems to adopt a more holistic approach, conceptualising biological maturation and relative age within a broader field of “push and pull factors” that impact challenge dynamics in talent development in sport. Finally, we provide practical guidance for talent systems in their approach to relative age and biological maturation, recognising that there is no “gold standard”. Instead, there is a need to recognize the highly individual and contextual nature of these concepts, focusing on strategic coherence through talent systems for the management of selection and development processes.

## 1. Introduction

In the competitive landscape of high-performance sport, there is significant pressure for talent systems to select and develop athletes to the senior elite standard [1]. On this basis, how limited resources are strategically used by talent systems has become a key issue in practice [2]. Reflecting this, thousands of young athletes across sports are selected to engage in often well-resourced development systems. The selection of young athletes into such programmes often occurs at young ages, and in sports like soccer, for example, can take place from as young as five years of age [3]. Those selected receive professional coaching and sports science and medical support, access to superior training equipment and facilities, and exposure to increased levels of competitive challenge when compared to non-selected peers [4,5]. The selection of the highest potential athletes into such a development programme is proposed to facilitate their long-term progress and increase probability of senior success [6]. Conversely, athletes who are not considered to show sufficient sporting promise at the time of selection are not recruited into these selective pathways and are denied access to such opportunities.

Recently, the means by which talent systems focus their resources has come under increasing scrutiny, with data challenging the established paradigms of talent development (TD) [6,7]. Of significant and ongoing debate is the timing of and access to selection, along with the way athletes are developed. Indeed, the predictive accuracy of early selection remains low, and even the best performing young athletes often fail to attain elite senior status [6]. Put simply, maximising efficiency through the early identification of athletes may come at the cost of effective practice as talent systems fail to invest resources in appropriate ways [1].

Two factors that have been examined in depth by the extant literature as influencing selection and development dynamics are biological maturation and relative age, e.g., [4,8,9,10,11,12,13,14]. Whilst the relative age effect (RAE) and biological maturation are often incorrectly interpreted as synonymous, more recent literature has emphasised that relative age and biological maturation are independent and individualised concepts [10,11,12,13]. Indeed, a recent qualitative study suggested that the RAE may be a population-level consequence of a constellation of factors less measurable than maturation alone [14]. Given the impact of relative age and biological maturation on the psychosocial development of young athletes, a key practical question across talent development systems and contexts is how these dynamics should influence practice. Traditionally, in respect to relative age and biological maturation, research and practice has tended to focus on the relative make-up of selection cohorts within TD systems and the impact of each concept on current performance status. Notably lacking, however, are discussions surrounding how these two concepts can be contextualised within the range of complex biopsychosocial factors that impact long-term development at the individual level. 

Reflecting these limitations, in this review, we aim to contextualise relative age and biological maturation more broadly in TD systems and subsequently offer ways in which talent systems may choose to engage in challenge management strategies. In the TD context, a developmental challenge is an experience perceived by a performer to have the potential of disrupting development and/or performance in sport [15]. Challenge dynamics are, therefore, the complex biopsychosocial factors that influence an individual’s experience of and interaction with challenge [16]. We begin this review by summarising biological maturation and relative age, then consider the various strategies that have been suggested to ‘counter’ their effects, before considering the broader range of challenge factors in development. We conclude by suggesting ways forward for talent systems regarding the management of these concepts. 

There is a significant gender bias in TD research [17], particularly in relation to the RAE and biological maturation. Reflecting on the disproportionate lack of research on female athletes, the differential male/female dynamics (e.g., physiological changes resulting from biological maturation, traditional ages of the onset of puberty, anthropometric profiles [18,19]) and potential differences in recommendations, our discussion in this review is delimited to male athletes. 

### 1.1. The Relative Age Effect

Relative age represents chronological age relative to the individual birthdate and competition cut-off dates [12]. The RAE is a selection bias in favour of those born earlier in the selection year, whereby those born toward the start of the selection year, who are chronologically older than those born toward the end of the selection year, are overrepresented within talent systems [4,12]. The RAE has attracted significant research attention and has been shown to exist across contexts and sports, with athletes born in the first two quartiles of the year disproportionately overrepresented at the expense of those born in the third and fourth quartiles [9,12,20,21,22,23,24,25]. For example, players born in the first quartile have been shown to constitute 56% of some soccer academy cohorts, with players born in quartile four comprising just 10% [9]. The literature has proposed that the multitude of attributes influencing the RAE are primarily related to age, experience, and developmental differences (e.g., game knowledge and understanding, decision making, neuromuscular development, cognition, behavioural and psychological development, social development) [12,25]. In youth sport, the RAE is present from early childhood and remains relatively stable throughout adolescent selection cohorts [10,11,12]. To this point, much of the RAE literature has emphasised the negative effects of chronological age groupings at the point of selection (e.g., [26,27]). Indeed, the consensus has been that there is the need to eradicate the RAE through developmental interventions [28] to prevent large numbers of young athletes from being excluded from talent systems [8]. Consequently, comparatively limited attention has been paid to the significant dropout of athletes in later selection cohorts [14] and the theoretical base underpinning the RAE [29].

### 1.2. Biological Maturation

Another factor influencing early advantage and selection is biological maturation, which, importantly, is a distinct construct to the RAE [10,11,12,13]. Biological maturation is the process of progression toward the mature adult state and can be defined in terms of status (the stage of maturation that the individual has attained at the time of observation), timing (the chronological age at which specific maturation events occur), and tempo (the rate at which maturation progresses) [30,31]. Of relevance to selection and development dynamics, children of the same chronological age can vary substantially in maturation status, timing, and tempo [4,32]. For instance, children of the same chronological age can vary by as much as five-to-six years in skeletal age, an established index of maturation status in youth [33,34,35]. Early maturation elicits numerous physiological, physical, and functional advantages (e.g., increased lean muscle mass, ability to reach faster peak speeds and perform more high-speed running, increased muscular strength and power) that transfer directly into performance environments [35,36,37,38,39,40,41]. Earlier maturation also generally confers greater body stature and mass [36,40]. These factors provide early maturing athletes with an advantage over peers and increases the likelihood of selection in contexts where these attributes are desirable. From the onset of puberty, biological maturation seems to have a stronger influence on selection than relative age in such contexts [10,11,12,13]. For instance, early biologically maturing athletes have been shown to constitute as many as 72% of youth soccer cohorts [11]. Late maturing athletes are frequently underrepresented, and in some instances, are absent from TD systems by age 14–15 years [11,12]. Reflecting this finding, TD practitioners and stakeholders have expressed concerns over the extent to which biological maturation influences selection and development in talent systems [42,43]. In addition, early maturation may confer enhanced self-efficacy and social status, alongside physical and functional performance advantages. Yet, if these advantages dissipate later, there may be maladaptive consequences for early maturing athletes when exposed to higher challenge levels at later stages of the pathway [44]. Contrastingly, if later maturing athletes lack the ability to cope with chronically low levels of early success, the likelihood of those athletes dropping out of the system is increased [45].

Importantly, maturation-related advantages are context-dependent; for example, in sports where prepubescent attributes are desirable for successful performance, such as some gymnastic events, delayed maturation may be advantageous for early performance and selection [30]. Maturation-related selection advantages are also influenced by other factors including the level of competition and even playing position in youth soccer [11,46]. Similar to relative age, the majority of research on biological maturation in respect to talent identification has tended to focus on the associated early selection advantages, with findings highlighting that the overrepresentation of early maturing youth within talent systems (i.e., soccer academies) emerges at the onset of puberty and increases in magnitude with chronological age and the level of competition [10,11,12]. Like the RAE, maturation-related selection biases are generally viewed as something to eradicate as a means of widening developmental opportunities for later maturing athletes. The desire for eradication, we would argue, does not have as simple a solution.

## 2. Interventions Targeted at Equalising Selection

Building on this assumption, the RAE and maturation biases have been seen as representing systemic selection error and, therefore, something to solve. Multiple interventions have been suggested to “level the playing field” and counteract RAEs and maturation-related selection biases in youth sports.

### 2.1. Selection Interventions Aimed at RAE

The first category of interventions predominantly views the RAE as the result of selection error, with a disproportionate population of relatively older athletes being given opportunities in talent systems. One intervention proposed to resolve this is age-ordered shirt numbering [27]. This intervention requires the number on the back of each player’s shirt to correspond with their order of relative age (in soccer, for example, the oldest player would wear number one and the youngest would wear number eleven) so that the ascending relative age order of each player is explicitly displayed to coaches during match-play, training, and assessments. Although suggested to eliminate the selection biases associated with the RAE in youth soccer [27], research is yet to be conducted to provide evidence to support the long-term validity or efficiency of age-ordered shirt numbering concerning either talent identification or development. By making relative age the focal point in the selection process, this strategy also seems to incorrectly view relative age as conferring universal advantage or disadvantage. Indeed, Mann and Van Ginneken [27] state that the intervention:

“May help coaches to provide more age-appropriate coaching and instructions so that it is tailored to each player’s expected skill level based on their age and/or maturation. Second, the age ordered shirt numbering could even help to make the individual players more aware of the differences in skill that should be expected as a result of their relative differences in age”(p. 788)

We suggest that if deployed in this manner, the assumption is ill-founded. In reality, those born at the start of the selection year can still be significantly disadvantaged relative to peers based on other factors [10,11,12,13]. Put simply, challenge dynamics cannot be assumed on the basis of a single variable [10,11,12,13]. Moreover, despite being advocated as an intervention that also influences biological maturation [27], the RAE and maturation are two independent constructs and an intervention targeted at mitigating one will not have a direct impact on the other [10,11,12,13]. It is also important to acknowledge that the inter-individual differences in relative age between players (a non-linear ascending function) cannot be accounted for using the linear ascending function of shirt numbers [27].

From a statistical standpoint, Cobley et al. designed a corrective adjustment procedure using longitudinal reference data from swimming performance metrics that were shown to remove the RAE in Australian state- and national-level swimmers [47]. Such corrective adjustments were calculated by generating accurate estimates of the relationship between decimal age and swimming performance based on repeated years of longitudinal cross-sectional performance data as a reference. When correctively adjusted swim times were examined, RAEs were absent across age-group and selection levels. Similar corrective adjustments have been utilised in athletics, with the suggestion that pre-existing RAEs can be effectively removed from all performance levels with such formulae [48]. Although an interesting proposition, we propose that (even if validated, effective and supported with longitudinal data) such a method would be limited to centimetres, grams, and seconds (CGS) sports (e.g., running, swimming, cycling) and is likely ineffective in sports where there are broader internal and external influences on performance outcomes (i.e., team sports, racquet sports). This is not to suggest that relative age strongly influences physical and functional performance in youth athletes, but rather that it is more difficult to control for the broader variety of confounding factors that influence performance outcomes in team and racquet sports, as opposed to the comparatively fewer in CGS sports. If such a corrective adjustment procedure were to be considered, it would be important to first identify the associations between relative age and performance in a given context before correcting for them. In addition, the second-order effects of levelling the challenge landscape are unknown, especially as challenge dynamics are experienced at the individual level and periods of high and low challenge appear desirable [43,45].

Several other interventions have been put forward to counteract RAEs. The establishment of quotas, where talent systems are required to select a minimum number of athletes from each birth quartile, has been suggested [49,50]. Another similar approach is an average age team rule, where the average age of the team is one-half of the age group range [51]. To date, however, and reflecting general limitations in TD research, there is a paucity of longitudinal data to support the impact of these inventions both on selection and, perhaps more importantly, on long-term development. By implementing these approaches, the RAE will be reduced and the number of athletes from each birth quartile will be more evenly distributed. Yet, a consequence of enforcing selection based purely on date of birth is that it removes the flexibility for coaches to make selection and deselection decisions based upon other biopsychosocial factors. Moreover, these structural approaches may remove the flexibility for the individual approach that might be required based on the unique set of circumstances in which an athlete finds themselves (e.g., play up or down). Indeed, all these approaches suffer from the assumption that the cause for disproportionate selection cohorts is a result of biased decision making. This perspective is focused on talent “identification” and the need to select the “right” people [52] but resultantly misses the broader picture of developmental dynamics that influence the route to selection and beyond [14]. It may be based on this perspective that no interventions have been sought to address the significant deselection of early born athletes, compared to their relatively younger peers [53,54].

### 2.2. Selection Interventions Aimed at Biological Maturation

Similar to the age-ordered shirt numbering intervention proposed to mitigate the selection biases associated with the RAE [27], player labelling has been suggested as a solution to overcome the selection biases associated with advanced biological maturation [55]. Player labelling requires the number on the back of each player’s shirt to correspond to the ascending order of players by maturity status. In practicality, during soccer training or competition, the most mature player would wear number one and the least mature player would wear number eleven so that coaches and scouts are aware of the variations in maturation status between players. When player labelling has been adopted in Swiss youth soccer, scouts at the regional level were shown to be less likely to rank the more biologically mature players as those with the most potential and, instead, were more likely to select the less mature players [55]. Crucially, however, when player labelling was not adopted (e.g., coaches were not provided maturation details of the players), there was no maturation selection bias in favour of either population [55]. By making biological maturation the sole focus in selection processes, and by incorrectly perceiving maturation status as conferring universal advantage or disadvantage, this intervention may create selection biases in favour of late maturing players based upon one single variable. Much like age-ordered shirt numbering, player labelling is also still in its relative infancy and no research has been conducted to produce findings to support the long-term validity or efficiency of the intervention concerning selection or development. Moreover, the inter-individual differences in maturation status between players (a non-linear ascending function) cannot be accounted for using the linear ascending function of shirt numbers [55]. Although providing coaches with visual cues to indicate the individual maturation statuses of the athletes within their care provides a progressive step forward from pre-existing methods, it is likely that the utility of the intervention will remain limited without the provision of coach education within this domain [11]. Ongoing educational support for practitioners in growth and maturation would help to support staff to support individual players based upon their physical needs and strengthen the utility of such interventions [46].

## 3. Interventions Targeted at Levelling the Developmental Playing Field

Although with some overlap with selection interventions and depending on how strategies are deployed, a second broad group of strategies have been designed to address not only selection biases but also the developmental dynamics experienced by athletes.

### 3.1. Developmental Interventions Focused on Relative Age

The second block of interventions have centred on presenting athletes with varying challenge levels and the opportunity to compete in different chronological age bandings. One such strategy is Birthday Banding, which aims to provide a range of developmental experiences in training and competition, where athletes move up to their next birthdate group on their birthday to remove fixed selection points and chronological age groups [56]. Under these conditions, an athlete can experience being the relatively youngest and oldest over a year, something that is proposed to confer a more diverse developmental experience. In one study, Birthday Banding was found to contribute to an insignificant RAE in a national squash TD system [56]. Although this is unlikely to be *the* single cause, it does suggest that this strategy can have a significant impact on the challenge level across a population. However, given the nature of challenge dynamics [57], and the advantages that variations in challenge levels offer, e.g., [45], Birthday Banding may remove the flexibility that might be required for the individual. For example, although some players may be of the chronological age to move up an age group, they may not have the psychosocial maturity or technical-tactical competency to cope with the challenge of the higher age group. In this sense, if such a policy is to be effective, flexibility within a TD system is required to allow coaches and practitioners to account for the individual developmental needs of each child and make such decisions (e.g., keep a player down an age group, move a player up before their birthday) on an individual basis. In addition, there may be maladaptive consequences for the application of Birthday Banding in team sports with significant turnover of groups and the consequent challenges of coherence and social dynamics. On the other hand, Birthday Banding may present one relatively low-resource intervention to provide fluctuations in challenge levels in individual sports. Given that it appears desirable for periods of both high and low challenge to be pulsed through a pathway [45], Birthday Banding may offer a window into how challenge dynamics might be manipulated.

A comparable strategy is the rotation of selection cut-off dates to reduce the number of relatively older athletes selected into the TD system [58]. Similarly, Hurley et al. proposed the Relative Age Fair Cycle, in which the cut-off dates for each year of competition are changed by three months between seasons of competition so that athletes experience being in all four quartiles of the year throughout development [59]. The advantage of this approach again seems to be the variety in competitive level faced by athletes. Yet, the implementation of such interventions seems to pose a variety of complex problems, requiring significant restructuring and potentially hindering coherence.

Another proposition to counter the RAE has been to delay selection until 15–16 years of age [8]. Critically, this proposition fails to take account of the dynamics that are at play regardless of selection into a talent system and how these might impact development. Indeed, there are National Governing Bodies that do not begin selection processes until these ages, and the RAE is still present in these systems upon selection, e.g., [53]. Crucially, removing the provision of high-quality TD processes until later stages of development (e.g., high-quality coaching, increased contact hours, and periodised challenge) may have detrimental effects on long-term development [45,60]. The likelihood of many TD systems (e.g., soccer academies) delaying selection until late adolescence is also very unlikely given the socio-political realities of professional sport [60]. In short, assuming that it is desirable to level the relative age playing field, macro strategies can only be part of the approach.

### 3.2. Developmental Interventions Focused on Biological Maturation

A variety of approaches have been suggested to counter the distinct advantages conferred by advanced biological maturation. Bio-banding is the most common and frequently investigated intervention to counteract the selection and performance advantages associated with variations in biological maturation. Bio-banding is proposed as an adjunct to, and not a replacement for, age group competition, forming just one part of a diverse developmental approach [31,61]. Bio-banding involves grouping and/or evaluating athletes based on maturity status rather than chronological age [31]. It is designed to promote competitive equity and athlete safety by limiting maturity-related variation in size and athleticism [31]. There are a number of proposed ways to bio-band athletes, including grouping them into maturity bands based upon their percentage of predicted adult height at the time of observation (e.g., 80–85%; 86–90%; 91–95% predicted adult stature [62]) and determining maturity offset (estimating the number of years athletes are from undergoing peak height velocity [63]).

There are several factors to consider regarding the evidence of the impact of bio-banding. Firstly, it is important to note that most research on bio-banding has tended to be soccer-specific and more research is required to understand its utility across a broader range of sporting contexts [31]. However, at specific time points, bio-banding has been perceived by some youth soccer players to generate greater physical, tactical, and technical challenges [32,41]. Early maturing players have described bio-banded competition as more physically challenging, reducing size and strength dependence and placing more emphasis on technical and tactical characteristics [41]. Conversely, later maturing players describe experiencing a greater opportunity to use, develop, and demonstrate their physical, social, technical, and psychological competencies in a less physically challenging environment [32,41]. Increased expression of technical-tactical characteristics because of bio-banded soccer competition/training has been found elsewhere, i.e., [64,65,66]. Moreover, bio-banding is perceived by some players of varying maturity statuses to promote differential social dynamics (e.g., leadership opportunities) [32,67]. Older children can benefit from taking up these teaching and leadership roles during bio-banded competition and may not get such opportunities in their own chronological age groups [68].

Bio-banded training camps have been introduced in sports such as cricket, with players again describing differential social and challenge dynamics [69]. Bio-banding has also been favourably received by various stakeholders in a youth soccer academy in relation to its impact on the psychological, social, and technical-tactical characteristics of the later maturing athletes [67]. Given that the risk of injury is influenced by maturation status (i.e., pre-, mid-, and post-peak height velocity) [31], bio-banding may also offer a method to prescribe maturity-specific training loads which may reduce injury risk and optimise conditioning effects in adolescent athletes [31]. However, in a recent commentary, Towlson and Cumming [61] argue that further research is required to determine when and how to best adjust training programmes to mitigate the risk of specific injuries and how this varies relative to the distal-to-proximal growth gradient.

Despite these positive findings, and whilst acknowledging that biological maturation clearly has a significant effect on challenge dynamics, it is important to note that bio-banding is not designed or expected to consider technical, tactical, cognitive, emotional, or social development [31,70]. For optimal development, there is a need to consider not just size- and maturity-related characteristics when grouping athletes by maturation status but also the plethora of complex factors, and their potential interactions, that influence individual development [31,70]. For this reason, in previous bio-banded competition events in youth soccer, participating teams were asked to consider each player’s psychological and technical-tactical competencies and to consider the exclusion of those individual players of the desired maturity statuses who may not benefit from bio-banded competition [41]. For instance, some athletes advanced in maturation may be capable of withstanding the physical demands of competing with chronologically older athletes, but they may not be of the required technical-tactical standard or psycho-social skills to cope [31,71]. As an example, whilst some athletes perceive bio-banding to provide the opportunity to make new friends across different age groups [41], other young athletes have previously reported feelings of apprehension brought about by the potential for social isolation when moving between different groups and not being with friends or other players whom they were familiar with [67]. Feelings of apprehension should in no way be considered as a universal negative and may be highly appropriate for some. Therefore, whilst there are examples of how TD systems have twinned bio-banding with offered psychological provision during periods of bio-banding [68], regardless of support offered, what may be appropriate for one athlete will not be for another. There is, therefore, a necessity to weigh up complex individual biopsychosocial factors to decide what is appropriate.

Furthermore, whilst some authors have quantitatively shown seemingly positive effects of bio-banding on aspects of technical-tactical performance in soccer (i.e., increased number of short passing sequences, reduced number of long passes [65]), others have observed a more limited effect on technical-tactical characteristics of players during bio-banded competition [72]. In addition, research from Spanish soccer academies has shown that matching players by maturity status alone in small-sided games formats elicited no skill differences displayed between groups [73]. This would seem to reiterate that biological maturation has no direct influence on technical skill levels and any intervention that focuses on grouping athletes solely by biological maturation will fail to account for such individual differences. On the other hand, using smaller training areas and pitch sizes presents one viable option to limit the extent to which earlier maturing athletes (e.g., soccer players, rugby players) can utilise their athletic advantages at the expense of other performance elements (e.g., technical-tactical characteristics) [41,73]. In this regard, Cumming et al. suggest a “hybrid approach” to bio-banding, consisting of monthly or bi-monthly bio-banded competitions, alongside existing games programmes [31].

Reflecting arguments offered earlier about RAE interventions, adopting blanket and routine bio-banding may fail to recognise the individualised and biopsychosocial nature of TD [31,74]. For example, a late maturing athlete that is relatively advantaged based on other factors (e.g., technical and tactical ability and social skills) may not benefit from competitive challenge being reduced even further [31]. Taken as a simplified example, if such steps are taken purely to level the playing field, this reduction in challenge may act as a barrier to long-term development [75,76,77]. Yet, if the intervention has other foci, such as the development of low-level psycho-behavioural skills, then it may be perfectly appropriate. In essence, this is a highly individualised matter, and we need to consider the intended impacts against the needs of the individual [78].

In addition to the challenges of individual dynamics, whilst acknowledging that the tracking of maturation is an essential feature of a talent system, the implementation of bio-banding can be resource intensive, both in terms of administration (e.g., the necessity of changing groups) and coaches understanding a wider range of individual needs. For many sports, whilst maturation testing itself can be reasonably simple depending on the method employed, it does require practitioners to conduct reliable measurements and track these longitudinally. Providing large populations of athletes with bio-banded training and competition opportunities at all stages of a talent system presents a significant challenge. It is also important to acknowledge that due to resource limitations (e.g., reduced access to skeletal X-ray assessments), coincided with the invasive nature of other predictive equations, non-invasive predictive equations to determine maturity status are commonly utilised (i.e., percentage of predicted adult height [62] or predicted maturity offset [63]). Due to the non-invasive and predictive nature of these equations, and as with all predictive equations, these methods are associated with a degree of error (e.g., [63,79,80,81]). Whilst the median error bounds between actual and predicted adult height using the Khamis-Roche method is just 2.2 cm in males aged between 4 to 17.5 years [62], this predictive equation is derived from retrospective datasets of American youth of European ancestry, and this must be acknowledged when applied to populations of differing nationalities and ethnicities. Moreover, both the updated and original equations for the predictive maturity offset method are suggested to be unreliable for both early- and late-maturing males and females, with an overestimation of the predicted ages at peak height velocity in early-maturing youth and an underestimation in late-maturing youth [81]. This inability to differentiate between early- and late-maturing youth using the maturity offset equation can lead to athletes being categorised incorrectly.

A suggested complement to bio-banding is Discreet Performance Banding (DPB), where athletes are grouped based on the performance of a discreet skill or ability that is highly valuable in their sport (e.g., change of direction ability in soccer), rather than using a marker of implied performance (e.g., maturation alone) [73]. DPB using change of direction ability in youth soccer has been suggested to differentiate variations in skill levels (passing, shooting, ball control), with the suggestion that it may hold the potential to level competition in youth sport from a skill perspective [73]. Therefore, it has been proposed that bio-banding, alongside DPB and chronological age group competition, may diversify the experiences of young athletes and expose them to new and varied challenges. However, the validity of a single discreet marker to differentiate between athletes seems highly questionable. We would suggest that it presents an overly blunt instrument that fails to take account of the broader biopsychosocial influences on performance. As an example, change of direction ability is a poor proxy for technical ability, tactical understanding, or psychological skills. This is especially the case as research on DPB is in its infancy and the method remains largely conceptual and untested [73].

Although not specifically an intervention for biological maturation or relative age, “playing up” athletes who have early advantages against chronologically older or higher-performing peers has been suggested to facilitate more appropriate levels of challenge and individual development [26]. Playing up has been perceived by youth soccer players to elicit improvements in fitness and sport-specific skill, social capital, and social adaptability, as well as being rewarding when recognition and success are experienced [82]. Indeed, these findings are somewhat unsurprising given the social status conferred by selection. Whilst “playing up” at face value may present athletes with a higher level of challenge, conferring some technical-tactical benefits [26], it may also lead to individuals relying on previously developed strengths, rather than developing potentially career-limiting weaknesses [44]. Athletes who play up may also face difficulties in coping with the increased intensity of competition and when fitting in with older teammates [82]. Somewhat counter intuitively, playing up can also provide a level of validation and reduced performance expectation that may actually reduce the perception of challenge [43]. Therefore, whilst it is clear that playing up or down significantly affects the perception of challenge, qualitatively, its impact cannot be assumed, with effects depending on a range of individual and environmental factors. In this sense, further qualitative and longitudinal research is required to understand the experiences of those who play up and the long-term benefits of developing expertise [26]. As with previous interventions, the application of these approaches is likely to be a highly individual matter.

## 4. Challenge Dynamics

The range of developmental interventions reviewed in this paper are aimed at the management of challenge dynamics through a pathway. To build a case for practical approaches it is, therefore, important to contextualise these dynamics within the existing literature. The importance of developing a range of psycho-behavioural skills to learn from and cope with challenges is well-established as an important requisite for developing excellence in sport [43,44,45,75,76,77,83,84,85,86]. Crucially, without the early acquisition and development of an adequate psychological skillset (that challenges can generate), athletes *can be* derailed by step changes in challenge that can occur towards the higher echelons of performance [43,44]. In contrast, if athletes develop psycho-behavioural skills, subsequently tested by a range of appropriate challenges, the consequent emotional disturbance, when coupled with appropriate support, can provoke further refinement of these skills [45].

Following this line of research, literature has challenged the assumption that being relatively younger or biologically late-maturing is unequivocally detrimental to development, instead identifying the potential later advantages of early disadvantage, e.g., [4,53,87,88,89]. One example is the reversal of relative age advantage, where relatively younger athletes are proportionately more likely to reach elite senior status despite a disproportionate number of relatively older athletes being selected at the youth level [7]. This is something now replicated across sporting contexts, e.g., [53,54]. Importantly, this is a reversal of advantage rather than the RAE reversing, suggesting that relatively younger athletes are less likely to be deselected than their relatively older counterparts [7]. Various mechanisms have been suggested to explain advantage reversals, including that relatively younger athletes are thought to benefit from the increased levels of competitive challenge when competing with their older counterparts who possess age- and experience-associated advantages (e.g., superior game knowledge and understanding). This increased level of competitive challenge has been proposed to benefit the relatively younger athletes, stimulating their adaptive development, and facilitating long-term progress. This has specifically been referred to as the “underdog hypothesis” [87]. Similarly, the comparatively greater challenge that is experienced by later maturing athletes within a development environment where they are competing against early-maturing athletes with physical, physiological, and functional advantages has been proposed to encourage the development of superior technical-tactical and psychological skills [4]. The development of these superior technical-tactical and psychological skills is proposed to allow the later-maturing athletes to survive and thrive in an environment where they are physically disadvantaged [4,87]. Although these superior technical, tactical, and psychological attributes may be less obvious throughout childhood and early adolescence, they are proposed to become salient in late adolescence and early adulthood once the physical advantages associated with advanced biological maturity become attenuated [4]. However, it is possible that many younger/late-maturing athletes always possessed such superior abilities which has allowed them to be initially selected into and remain within the system. Indeed, it is equally plausible that many early-maturing players also possess and/or develop superior technical-tactical and psychological skills within the same TD system despite not being exposed to the same physical challenges [90].

There is some evidence to suggest that late-maturing soccer and rugby academy players and relatively younger rugby players are proportionately more likely to progress to the elite adult level than early-maturing/relatively older players if retained within the system [53,54,88,89]. Whilst in support of the underdog hypothesis, it is also important to recognise the opposing methods used to estimate/examine maturity status (i.e., maturity offset method [63] vs. TW3 method [91]) within these biological maturation-specific investigations, as well as the different criteria used to classify early, late, and on-time athletes [88,89]. Contrasting evidence from Swiss national-level youth soccer also suggests that many late-maturing players, despite possessing superior technical abilities and being exposed to the “underdog challenges”, are still deselected from the TD system by age 15 years [90]. However, this does not suggest that these players still did not progress to become elite senior athletes; there is no longitudinal data to indicate how these challenges influenced long-term development through to the senior level [90]. Critically, however, if the underdog hypothesis were to exist, potential “underdog” effects of being relatively younger or biologically late-maturing would only hold if relatively younger/late-maturing athletes are retained within the system.

Contrasting with the original underdog hypothesis [87], it is not the provision of higher challenge but, instead, how the individual responds to challenge that is a key determinant of success [15,77]. Rather than directly causing development, challenge acts to test previously developed psycho-behavioural skills [85,92]. As we have previously discussed [11], it is important to note that late-maturing players likely remain underrepresented at the adult level in absolute terms due to a smaller initial representation within the academy system. Indeed, a prime example of this in a relative age context has been presented in U17-, U19-, and senior-level international male soccer players [54]. Reflecting general limitations in biological maturation and RAE research, there is a lack of longitudinal data to support this proposition across contexts. End-stage conversion rates are often used as a metric for the underdog hypothesis [87], but caution is advised when examining end-stage conversion rates as a metric for career outcomes as some populations *may* still be over- or underrepresented in absolute terms [11].

### Push and Pull Factors

Being born late in the selection year or being late-maturing biologically does not serve as a direct advantage or disadvantage; instead, the dynamics of challenge events are highly individual. Presenting a holistic view of challenge dynamics, McCarthy et al. suggested the concept of push and pull factors to conceptualise factors that may confer relative advantage or disadvantage for the athlete [14]. Their hypothesis is that at the population level, the average early-born athlete will be subject to more push factors—those factors that act to accelerate early performance—whereas the later-born athlete will be subject to more pull factors—factors that act to retard early performance. Abundant push factors will encourage early performance at the junior level but may hinder later progress. In contrast, those athletes who experience a greater prevalence of pull factors *may* experience a more developmentally optimal experience [75]. Importantly, however, those athletes who are subject to an overwhelming volume of “pull” factors might also become chronically disadvantaged, especially if these are external to the sport [93]. This seems to support the notion that significant challenge factors in an individual’s life outside of sport are not an adaptive feature of development [76,77]. There is, therefore, a practical necessity for strategic consideration and a subtle balance in how this approach is applied. An overabundance of pull factors may risk derailment, with repeated performance setbacks, negative feedback, and resultant negative emotional states unlikely to build an athlete’s motivational resources [44,45,94]. In essence, pull factors are not necessarily positive for overall development, especially if preventing athletes from ever being selected (e.g., [11,12]). In the context of the literature presented earlier in this paper, later-born athletes will, on average, be subject to more pull factors. As a result, the typical later-born selected athletes will be provided with a higher frequency and intensity of challenge to navigate as they progress. Reflecting the distinction between relative age and maturation, this suggests that early maturation is, in itself, an independent push factor, whereas late maturation is a pull factor. Thus, early-maturing athletes are more likely to have significant physical, physiological and functional advantages and consequently are more likely to be selected in contexts where these attributes provide advantages relative to peers. However, competing against less-mature peers may prevent the testing of psychological skills for the early-maturing athletes [4,41]. Indeed, late biological maturation appears to be one of the most prominent pull factors in the TD context (e.g., [10,11,12,46]).

Based on the review of research presented, which consistently identifies early selection advantages based on push factors, a narrative in TD practice has been the desire to “level the playing field” and prevent inequalities of outcome in talent systems. These discussions are symptomatic of what could be framed as a “wicked” problem: one with no ultimate solution or stopping point, where better or worse is a value judgement based on desired state [95]. Reflecting these complex dynamics and our review of existing approaches, we will now address the second aim of this paper by presenting considerations for talent systems seeking to utilise challenge dynamics in an evidence informed manner [96].

## 5. Implications for Practice in Talent Systems

Seeking to present implications for practice, we would suggest a need to conceptualise the interventions that are critiqued within the broader whole. In doing so, we suggest that push and pull factors may offer the potential for a holistic and practical view of challenge in talent systems in sport. For those seeking to operationalise these factors, there is the need for a highly context-dependent and, ideally, individualised approach. Talent systems can be considered at three levels. The macro represents the interactions between organisations, typically at the national/international level (e.g., NSOs). The meso is typically a collection of microsystems or ‘all aspects of the coaching situation’ [97] (p. 345) (e.g., an academy). The micro level represents the individual interactions that occur day-to-day in practice [2]. Thus, rather than looking at single variables, there is a need for talent systems to frame their actions in a deeper understanding of the effects and potential side effects of interventions. This requires maximum flexibility at each level of a system.

### 5.1. Micro-Level Implications

Reflecting the emphasis on individual dynamics, we refer first to the micro level. This contrasts with the majority of reviewed interventions, which predominantly aim at the macro or meso level. At the micro level, if individuals are to be presented with appropriate challenge levels, there is a necessity to adapt to inter- and intra-individual differences with a focus on the perceptions and needs of the individual athlete. This relies on a granular understanding of athletes’ circumstances and the empathic accuracy necessary to notice change, e.g., [98]. In all instances, an understanding of the maturational status of athletes is a useful data point for the coach and practitioner to make sense of the needs of a particular athlete. Yet, it should not be the only matter for consideration, nor should assumptions be made regarding relative advantage or disadvantage based on a single data point. Take the example presented earlier: a late-maturing athlete that holds a technical advantage relative to their age group is unlikely to be further challenged under the same circumstances playing against peers who are chronologically younger but matched in maturation [31]. However, if the goal was to expand their tactical understanding, develop leadership skills [41], or provide a metacognitive challenge [99], playing down an age group might facilitate the social circumstances necessary to support this process. Likewise, an early-maturing sprinter who has dominated in their chronological age group might strongly dislike the experience of competing against older, but more capable athletes. It may also be the experience that facilitates a refocus on weaknesses in their approach. Whilst competing at a higher level may seem like an appropriate intervention to provide additional levels of challenge, if an athlete does not have the psycho-behavioural competencies required to benefit, such a step change in challenge may be too great for that athlete to handle. Most importantly, in all these circumstances, it is not the event itself that will automatically confer development. Instead, it is the athlete’s perspective and use of psychological skills, actively shaped by coaches and peers, that is critical [100].

On a broader note, at the micro level, it is also important to understand the individual context of each athlete from the totality of their experience and their lives, rather than solely age or maturity status alone. Take, for example, an athlete that is facing an abundance of pull factors outside of sport; after all, maturation or relative age is not the only important consideration, particularly in instances where an athlete is overwhelmed by an abundance of pull factors outside of the athletic domain. This approach is something which, despite some debate, is increasingly acknowledged as undesirable for TD [77,93,101]. In essence, this becomes a more holistic and individual process, understanding a range of different push and pull factors and how they impact individual talent trajectories along with the ability of individual practitioners to make effective decisions about what is needed for those athletes. This will require a broader approach than, for example, playing up/down or finding alternate means of grouping athletes. It necessitates fundamental changes to practice, requiring the coach or practitioner to actively present individuals with appropriate levels of challenge. Reflecting our previous points in respect to bio-banding, this presents a necessity for a level of expertise to respond to individual developmental needs and experiences of each athlete as they arise [102]. In this sense, the Professional Judgement and Decision Making (PJDM) of the individual coach or TD practitioner becomes a key facilitator of learning and progress [78,103].

### 5.2. Meso-Level Implications

To enable these processes at the individual level, we offer several key considerations at the organisational level beyond those made previously in the literature regarding recommendations for effective practice (e.g., [78,104,105]). Specifically, we make suggestions for the operationalisation of push and pull factors in terms of selection and challenge management. The identification and selection of athletes has received significant attention in both the literature and practice, often being viewed as distinct from challenge dynamics. We suggest that rather than a search for athletes most likely to progress to the elite level, a more developmental and practical lens is applied to selection processes. This requires a focus not only on current performance, but also on the likelihood of further development [52] and contribution to the further development of peers. In this regard, it is likely that the previous successful navigation of sport-based challenges, such as a high prevalence of pull factors like late biological maturation, relative to current performance, may signal what Baker et al. referred to as “high potential” [52]. That is, despite being subject to significant challenge, an athlete is “sticking in there”. This raises several practical questions: how can we know about these factors if our approach to selection is the “talent scout” observing performance alone? Similarly, we cannot rely on single-variable interventions (e.g., [27]) used as a proxy for the complex web of dynamic challenge factors that seem to play a central role in development. Strategically this will require systems to hold a contextually defined view of the purpose and function of selection. At the meso level, this would see organisations (e.g., individual academies) moving away from traditional talent spotting, something that is consistently doubted in the literature [6,106], towards a more pragmatic, contextualised view with decisions made based on a broader picture of biopsychosocial factors [107]. For example, in some sports, early selection is a political necessity and one that affords the opportunity to shape a developmental journey [60]. In others, macro national systems have legislated to limit the timing of selection and for a broader population to be selected (e.g., English rugby union regional academies [53]), yet with less opportunity to shape athlete development over time. In other contexts, like Norwegian handball, selection is seen as supplementary, with participation, development and performance contexts running in parallel; an approach that promotes a breadth of engagement opportunities but presents additional challenges to the shaping of athlete experience [108,109]

Whilst we do not believe that there is a need for organisations to move towards a complete equity of push and pull factors, if selection processes at the organisational level exclude cohorts of athletes, this should warrant deliberate attention. This attention should include a pragmatic discussion regarding the relative weighting of resources needed to address the disparity. For example, data presented by Sweeney et al. suggest a near total exclusion of late-maturing soccer players in Ireland [11]. This is a significant issue, especially where there are limited routes back into the pathway, or where there is a marked difference in development provision between those who are in or out of the system. Yet, given that there is no optimal balance of what selection should look like (e.g., the proportion of early-, on-time-, and late-maturing players within an academy), organisations should be encouraged to critically reflect on the desirability of selection cohorts being strongly weighted towards push factors like early biological maturation [10,11,12].

Moving beyond the reasoning that the function of talent systems is simply to select the right athletes, the second core meso-level concern is the management of challenge throughout development. Many of the foci for proposed solutions suggest the benefit of an overall fluctuation of challenge level for the individual athlete, elsewhere referred to as periodised challenge [76]. Rather than relying on targeting single variables, at the organisational level, the focus should be placed on challenge management. The ideal output of this approach would be the integration of systems and support figures to maintain coherence for the athlete [43]. In high-resource organisations (e.g., category one soccer academies), this may be multiple staff feeding into a development plan for an athlete directing the types of experiences appropriate for their development, informed by an assessment of push and pull factors. At the lower resourcing end (e.g., smaller Olympic sports), this profiling might be done by an individual coach. Where a range of support figures are present (e.g., coaches, practitioners, parents), there is a need for a shared understanding of individual challenge dynamics, especially if athletes are to move between different levels of performance and training groups. It is for this reason that shared mental models (SMMs) have been proposed as a vehicle to support integrated practice [110]. SMMs refer to an organised and common understanding among team members regarding the essential aspects of work and how they should behave in specific situations [111]. SMMs among coaching and support staff would allow coaches to understand each athlete’s individual needs and adapt their decisions based on individual circumstances. Ongoing case conferencing, coaching communities of practice and review processes that are designed for the co-construction and sharing of knowledge amongst staff becomes essential, especially as SMMs cannot be assumed as a function of time spent together [112,113].

We should also recognise that such flexibility at the meso level poses a challenge to integrated practice, especially in resource-intensive systems with large numbers of staff where risks of a lack of vertical and horizontal coherence exist (e.g., [42]). For example, if a dominant early-maturing adolescent athlete is selected for a senior competition, we cannot assume this selection automatically confers a higher level of challenge. Instead, it is how the athlete’s experience is curated that matters. This requires multiple people to hold a shared understanding of the purpose and plan. It could be done in a manner that confers the athlete with enhanced self-efficacy or lower pressure based on role clarity [100]. In contrast, it could also be used to highlight weaknesses, generating feedback in areas that have previously not been challenged with and against age-matched peers.

Therefore, we suggest a necessity for organisations to monitor the prevalence of push and pull factors for their athletes on a longitudinal basis, utilising individual biological maturation along with other suggested push and pull factors that have been identified in the literature (e.g., familial influence [114], socio-economic status [115], and quality of previous coaching [116]). In contrast, we suggest that relative age data should be used differently, given that at the individual level it might not indicate individual challenge dynamics. Instead, it could be used to understand the relative make-up of selection cohorts over time, or to consider the efficacy of challenge management processes if higher proportions of players with greater early advantages continue to be deselected [7,53].

### 5.3. Macro-Level Implications

Building from the micro and meso, our main critique thus far of most existing and well-intentioned approaches to the management of challenge is the lack of individual flexibility of approach. We would suggest that any strategic approach to relative advantage or disadvantage should be focused at the meso and micro level, rather than at the macro level. Consequently, we suggest the necessity of a more fine-grained approach to the grouping of young athletes and the provision of challenge [14]. As noted by Cumming et al., if approaches like bio-banding are to be adopted wholesale, then we may simply advantage and disadvantage a different group based on less-measurable constellations of characteristics [31]. Implementing blanket strategies to mediate against disproportionately high pull factors is overly simplistic and lacks holistic consideration of the biopsychosocial factors that influence relative advantage or disadvantage. Many existing approaches also focus on attempting to “level out” challenge level which, based on the existing evidence base and applied at the population level, may be suboptimal given that it may be desirable for periods of high and low challenge levels to be pulsed through a pathway [45]. Indeed, talent systems may also ask if it is desirable for those who experience more push factors in development to be grouped together for appropriate challenge, so long as there is a vehicle for others to receive appropriately high-quality development.

Ultimately, rather than a bureaucratic regulatory approach, flexible systems should allow for practitioners to make decisions and respond to individual athlete needs. Therefore, we suggest the need for maximum flexibility and informed decision making for organisations and individuals. As is currently the case in the vast majority of systems, it may be easier for the top-down mandating of one approach for all, rather than encouraging informed flexibility. However, based on a model of a top-down *and* bottom-up approach to talent strategy [2], we suggest that as far as possible, national systems remove barriers to optimising individual challenge, as well as provide high-quality input to individual organisations to enhance their approaches. As an example, the Royal Belgian Football Association’s Futures Programme is a meso approach to provide developmental opportunities for later-maturing athletes [117]. It enables opportunities for late-maturing players to be retained within the system and experience training, competition, coaching, and travel as part of a national team. In this instance, the strategy still means that selection is based on players being identified as technically, tactically, and psychologically able for youth international soccer. However, if this approach was adopted as a policy requirement on a broader scale, it would prevent existing organisations from adopting strategies appropriate to their unique context.

In terms of “fairness”, it is here that macro systems characterised without a step change in the quality of environment between those who are selected and not seem to hold an advantage (e.g., [109]). Yet, this also means the provision of high-quality support to a large population which is highly resource-intensive [2]. A key feature of our position is that there are no value-free judgements to be made in this area. At all levels, we need to recognise the various trade-offs inherent to managing the dynamics of development. If the more holistic approach we suggest is to be adopted, there is also a need to promote decision making and integrated action through intelligent mediums. As such, any strategic approach should be enabled by the macro, but should be targeted at the meso and micro level of a TD system. This necessitates macro support for coach and practitioner development on a holistic and evidence informed basis. In addition, the need to generate SMMs of outcome and performance is likely a necessity, especially where multiple stakeholders impact on the curriculum of an athlete [107,108,109].

## 6. Conclusions

In this conceptual paper, we have reviewed the literature that seeks to negate some of the various selection and challenge dynamics in talent systems. We have suggested that whilst many of these biases may come at the population level, the dynamics of challenge effects are highly individual. In all cases, we suggest that research and practice view the use of challenge mitigation approaches, like bio-banding, as tools to use at the individual level rather than strategies to deploy at the macro or meso level. There is no “gold standard” approach to challenge management. What constitutes effective practice in this regard is highly contextual and determined by a myriad of other biopsychosocial factors that extend far beyond date of birth or current maturation status alone. As a consequence, whilst there is of course a need to understand the dynamics illustrated by the vast literature in biological maturation and relative age, there is also a need for the research to investigate less quantifiable factors that might impact development. In addition, we suggest the need for researchers to appreciate this broader and perhaps interdisciplinary picture, along with the value proposition of interventions in talent systems. Thus, a key recommendation in regard to challenge dynamics would be an end to the focus on “levelling the playing field” of a phenomenon that has so many complex factors at play. In practice, there is an opportunity for talent systems to adopt a more holistic approach by conceptualising biological maturation and relative age within a broader spectrum of challenge dynamics and considering how other, less-measurable factors also impact athlete development.

## Data Availability

Not applicable.

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
