# Peer review of "Push and Pull Factors: Contextualising Biological Maturation and Relative Age in Talent Development Systems"

_children, 2023, doi:10.3390/children10010130_

Round 1

Reviewer 1 Report

Please, see my comments in the document attached   

Author Response

Dear Reviewer, 

Thank you for your insightful comments provided in your review. We have responded to each comment directly within the document attached. Please also note that on the revised manuscript, all changes have been made in red coloured text for ease of review. 

Reviewer 2 Report

The data presented and discussed here is interesting, as analyse an overview of biological maturation and relative age in talent development systems.  

The authors did a great work reviewing and discussing the current topic. 

Thus, despite the interesting topic, I have only some questions that need to be more clear. 

1. L 14-16: " We suggest that the relative age effect may predominantly be a population level effect, whereas biological maturation acts at the individual level to impact the challenge level for individual athletes. "Please, explain better given more details. 

2. L 69-71: "Reflecting the 69 differential male/female dynamics [e.g., 15, 16] and potential differences in 70 recommendations, our discussion in this review is delimited to male athletes. " Please, explain better given more details. 

3. I suggest the authors to summarize some of the information in the text, using text boxes. For instance, for practical guidance for talent systems in managing relative age and biological maturation, authors should use tables and/or figures to summarize the information. I think it will be more easier and enriching for readers if we have some short key messages at the end of each heading. 

Author Response

(The authors gave the same response as above.)

Round 2

Reviewer 1 Report

Dear Authors,

I hope you are doing very well.

Great work on revising the paper, congratulations. I am happy with the current shape of this work, so I recommend its acceptance. 

Just a brief note: I knew that many of the papers shared on my previous revision will not fit directly with your work, I mean be appropriate to use as references, but I shared with you anyway because I believe that many of that concepts could be applied - reconceptualised - according to this field of research... 

Anyway, great job and great contibution to the field!

Wish you a wonderful 2023!

Kind regards, 

Author Response

Thank you to the reviewer for their insightful and informative review and their response to our revisions.